# WHEN IS OFFLINE POLICY SELECTION FEASIBLE FOR REINFORCEMENT LEARNING?

## ABSTRACT

Hyperparameter selection and algorithm selection are critical procedures before deploying reinforcement learning algorithms in real-world applications. However, algorithm-hyperparameter selection prior to deployment requires selecting policies offline without online execution, which is a significant challenge known as offline policy selection. As yet, there is little understanding about the fundamental limitations of the offline policy selection problem. To contribute to our understanding of this problem, in this paper, we investigate when sample efficient offline policy selection is possible. As off-policy policy evaluation (OPE) is a natural approach for policy selection, the sample complexity of offline policy selection is therefore upper-bounded by the number of samples needed to perform OPE. In addition, we prove that the sample complexity of offline policy selection is also lower-bounded by the sample complexity of OPE. These results imply not only that offline policy selection is effective when OPE is effective, but also that sample efficient policy selection is not possible without additional assumptions that make OPE effective. Moreover, we theoretically study the conditions under which offline policy selection using Fitted Q evaluation (FQE) and the Bellman error is sample efficient. We conclude with an empirical study comparing FQE and Bellman errors for offline policy selection.

## 1 INTRODUCTION

Learning a policy from offline datasets, known as *offline reinforcement learning (RL)*, has gained in popularity due to its practicality. Offline RL is useful for many real-world applications, as learning from online interaction may be expensive or dangerous (Levine et al., 2020). For example, training a stock-trading RL agent online may incur large losses before learning to perform well, and training self-driving cars in the real world may be too dangerous. Ideally we want to learn a good policy offline, before deployment.

In practice, offline RL algorithms often have hyperparameters which require careful tuning, and whether or not we can select effective hyperparameters is perhaps the most important consideration when comparing algorithms (Wu et al., 2019; Kumar et al., 2022). Additionally, performance can differ between algorithms, so algorithm selection is also important (Kumar et al., 2022). When a specific algorithm is run with a specific hyperparameter configuration, it outputs a learned policy and/or a value function. Thus, each algorithm-hyperparameter configuration produces a candidate policy from which we create a set of candidate policies. The problem of finding the best-performing policy from a set of candidate policies is called *policy selection*. While policy selection is typically used for algorithm-hyperparameter selection, it is more general since each candidate policy can be arbitrarily generated.

The typical way of finding the best-performing policy in RL is to perform several rollouts for each candidate policy in the environment, compute the average return for each policy, and then select the policy that produced the highest average return. This approach is often used in many industrial problems where A/B testing is available. However, this approach assumes that these rollouts can be performed, which necessarily requires access to the environment or simulator, a luxury not available in the offline setting. As such, in offline RL, Monte Carlo rollouts cannot be performed to select candidate policies, and mechanisms to select policies with offline data are needed.

A common approach to policy selection in the offline setting, known as *offline policy selection,* is to perform off-policy policy evaluation (OPE) to estimate the value of candidate policies from a fixed dataset and then select the policy with the highest estimated value. Typical OPE algorithms include direct methods such as fitted Q evaluation (FQE) estimator (Le et al., 2019), importance sampling (IS) estimator (Sutton & Barto, 2018), doubly robust estimator (Jiang & Li, 2016), model-based estimator (Mannor et al., 2007), and marginalized importance sampling estimator (Xie et al., 2019). Empirically, Tang & Wiens (2021); Paine et al. (2020) provide experimental results on offline policy selection using OPE. Similarly, Doroudi et al. (2017); Yang et al. (2022) use OPE estimators for policy selection.

Offline policy selection has been mainly associated with OPE, since these two problem are closely related. It is known that OPE is a hard problem that requires an exponential number of samples to evaluate any given policy in the worst case (Wang et al., 2020), so OPE can be unreliable for policy selection. As a result, a follow-up question is: do the hardness results from OPE also hold for offline policy selection? If the answer is yes, then we would need to consider additional assumptions to enable sample efficient offline policy selection.

Moreover, offline policy selection should be easier than OPE intuitively, since estimating each policy accurately might not be necessary for policy selection. There is mixed evidence that alternatives to OPE might be effective. Tang & Wiens (2021) show empirically that using TD errors perform poorly because they provide overestimates; they conclude OPE is necessary. On the other hand, Zhang & Jiang (2021) perform policy selection without OPE, by selecting the value function that is closest to the optimal value function. However, their method relies on having the optimal value function in the set of candidate value functions. It remains an open question about when, or even if, alternative approaches can outperform OPE for offline policy selection.

Unfortunately, there is little understanding about the offline policy selection problem to answer the aforementioned questions. Therefore, to provide a better understanding, we aim to investigate the question: *When can we perform offline policy selection efficiently (with a polynomial sample complexity) for RL?* To this end, our contributions are as follows:

- We show that the sample complexity of the offline policy selection problem is lower-bounded by the samples need to perform OPE. This implies no policy selection approach can be more sample efficient than OPE in the worst case. On the other hand, OPE can be used for offline policy selection, so the sample complexity of policy selection is upper-bounded by the samples required for OPE. In particular, we show that a selection algorithm that simply chooses the policy with the highest IS estimate achieves a nearly minimax optimal sample complexity, which is exponential in the horizon.

- To circumvent exponential sample complexity, we need to make additional assumptions. We identify when FQE, a commonly used OPE method, is efficient for offline policy selection. Specifically we discuss that FQE is efficient for policy selection when the candidate policies are well covered by the offline dataset. This theoretical result supports several empirical findings.

- We explore the use of Bellman errors for policy selection and provide a theoretical argument and experimental evidence for the improved sample efficiency of using Bellman errors compared to FQE, under stronger assumptions such as deterministic dynamics and good data coverage.

## 2 BACKGROUND

In reinforcement learning (RL), the agent-environment interaction can be formalized as a finite horizon finite Markov decision process (MDP) $M = (\mathcal{S}, \mathcal{A}, H, \nu, Q)$. $\mathcal{S}$ is a set of states with size $S = |\mathcal{S}|$, and $\mathcal{A}$ is a set of actions with size $A = |\mathcal{A}|$, $H \in \mathbb{Z}^+$ is the horizon, and $\nu \in \Delta(\mathcal{S})$ is the initial state distribution where $\Delta(\mathcal{S})$ is the set of probability distributions over $\mathcal{S}$. Without loss of generality, we assume that there is only one initial state $s_0$. The reward $R$ and next state $S'$ are sampled from $Q$, that is, $(R, S') \sim Q(\cdot|s, a)$. We assume the reward is bounded in $[0, r_{max}]$ almost surely so the total return of each episode is bounded in $[0, V_{max}]$ almost surely. The stochastic kernel $Q$ induces a transition probability $P : \mathcal{S} \times \mathcal{A} \to \Delta(\mathcal{S})$, and a mean reward function $r(s, a)$ which gives the mean reward when taking action $a$ in state $s$.

A non-stationary policy is a sequence of memoryless policies $(\pi_0, \ldots, \pi_{H-1})$ where $\pi_h : \mathcal{S} \to \Delta(\mathcal{A})$. We assume that the set of states reachable at time step $h$, $\mathcal{S}_h \subset \mathcal{S}$, are disjoint, without loss

of generality, because we can always define a new state space $\mathcal{S}' = \mathcal{S} \times \{0, 1, 2, \ldots, H-1\}$. Then, it is sufficient to consider stationary policies $\pi : \mathcal{S} \to \Delta(\mathcal{A})$.

Given a policy $\pi$, for any $h \in [H]$, and $(s, a) \in \mathcal{S} \times \mathcal{A}$, we define the value function and the action-value function as $v_h^\pi(s) := \mathbb{E}^\pi[\sum_{t=h}^{H-1} r(S_t, A_t)|S_h = s]$ and $q_h^\pi(s, a) := \mathbb{E}^\pi[\sum_{t=h}^{H-1} r(S_t, A_t)|S_h = s, A_h = a]$, respectively. The expectation is with respect to $\mathbb{P}^\pi$, which is the probability measure on the random element $(S_0, A_0, R_0, \ldots, R_{H-1})$ induced by the policy $\pi$. The optimal value function is defined by $v_h^*(s) := \sup_\pi v_h^\pi(s)$, and the Bellman operator is defined by

$$(\mathcal{T}q_h)(s, a) = r(s, a) + \sum_{s' \in \mathcal{S}} P(s, a, s') \max_{a' \in \mathcal{A}} q_h(s', a').$$

The Bellman evaluation operator is defined by

$$(\mathcal{T}^\pi q_h)(s, a) = r(s, a) + \sum_{s' \in \mathcal{S}} P(s, a, s') \sum_{a' \in \mathcal{A}} \pi(a'|s') q_h(s', a').$$

We use $J(\pi)$ to denote the value of the policy $\pi$, that is, the expected return from the initial state $J(\pi) = v^\pi(s_0)$. A policy $\pi$ is optimal if $J(\pi) = v^*(s_0)$.

In the offline RL setting, we are given a fixed set of transitions $D$ with samples drawn from a data distribution $\mu$. In this paper, we consider the setting where the data is collected by a behavior policy $\pi_b$ since the data collection scheme is more practical (Xiao et al., 2022) and is used to collect data for benchmark datasets (Fu et al., 2020). We use $d_h^\pi$ to denote the data distribution at the horizon $h$ by following the policy $\pi$, that is, $d_h^\pi(s, a) := \mathbb{P}^\pi(S_h = s, A_h = a)$, and $\mu_h(s, a) := \mathbb{P}^{\pi_b}(S_h = s, A_h = a)$.

**Notation.** We use $[H]$ to denote the set $[H] := \{0, 1, \ldots, H-1\}$. Given a value function $q$ and a state-action distribution $\mu$, the norm is defined as $\|q\|_{p,\mu} := (\sum_{s,a} \mu(s, a)|q(s, a)|^p)^{1/p}$ and the max norm is defined as $\|q\|_\infty := \max_{s,a} |q(s, a)|$.

## 3 SAMPLE COMPLEXITY OF OFFLINE POLICY SELECTION

We consider the offline policy selection (OPS) problem and offline policy evaluation (OPE) problem. We follow a similar notation and formulation used in Xiao et al. (2022) to formally describe these problem settings. The OPS problem for a fixed number of episodes $n$ is given by the tuple $(\mathcal{S}, \mathcal{A}, H, \nu, n, \mathcal{I})$. $\mathcal{I}$ is a set of instances of the form $(M, d_b, \Pi)$ where $M \in \mathcal{M}(\mathcal{S}, \mathcal{A}, H, \nu)$ specifies an MDP, $d_b$ is a distribution over a trajectory $(S_0, A_0, R_0, \ldots, R_{H-1})$ by running the behavior policy $\pi_b$ on $M$, and $\Pi$ is a finite set of candidate policies. We consider the setting where $\Pi$ has a small size and does not depend on $S$, $A$ or $H$.

An OPS algorithm takes as input a batch of data $D$, which contains $n$ trajectories, and a set of candidate policies $\Pi$, and outputs a policy $\pi \in \Pi$. We say an OPS algorithm $\mathcal{L}$ is $(\varepsilon, \delta)$-sound on instance $(M, d_b, \Pi)$ if

$$\Pr_{D \sim d_b}(J_M(\mathcal{L}(D, \Pi)) \geq J_M(\pi^\dagger) - \varepsilon) \geq 1 - \delta$$

where $\pi^\dagger$ is the best policy in $\Pi$. We say an OPS algorithm $\mathcal{L}$ is $(\varepsilon, \delta)$-sound on the problem $(\mathcal{S}, \mathcal{A}, H, \nu, n, \mathcal{I})$ if it is sound on any instance $(M, d_b, \Pi) \in \mathcal{I}$.

Given a pair $(\varepsilon, \delta)$, the sample complexity of OPS is the smallest integer $n$ such that there exists a behavior policy $\pi_b$ and an OPS algorithm $\mathcal{L}$ such that $\mathcal{L}$ is $(\varepsilon, \delta)$-sound on the OPS problem $(\mathcal{S}, \mathcal{A}, H, \nu, n, \mathcal{I}(\pi_b))$ where $\mathcal{I}(\pi_b)$ denotes the set of instances with data distribution $d_b$. That is, if the statistical complexity is lower-bounded by $n$, then, for any behavior policy $\pi_b$, there exists an MDP $M$ and a set of candidate policies $\Pi$ such that any $(\varepsilon, \delta)$-sound OPS algorithm on $(M, d_b, \Pi)$ requires at least $n$ samples.

Similarly, the OPE problem for a fixed number of episodes $n$ is given by $(\mathcal{S}, \mathcal{A}, H, \nu, n, \mathcal{I})$. $\mathcal{I}$ is a set of instances of the form $(M, d_b, \pi)$ where $M$ and $d_b$ are defined as above, and $\pi$ is a target policy. An OPE algorithm takes as input a batch of data $D$ and a target policy $\pi$, and outputs an estimate of the policy value. We say an OPE algorithm $\mathcal{L}$ is $(\varepsilon, \delta)$-sound on instance $(M, d_b, \pi)$ if

$$\Pr_{D \sim d_b}(|\mathcal{L}(D, \pi) - J_M(\pi)| \leq \varepsilon) \geq 1 - \delta.$$

We say an OPE algorithm $\mathcal{L}$ is $(\varepsilon, \delta)$-sound on the problem $(\mathcal{S}, \mathcal{A}, H, \nu, n, \mathcal{I})$ if it is sound on any instance $(M, d_b, \pi) \in \mathcal{I}$. Note that $\varepsilon$ should be less than $V_{max}/2$ otherwise the bound is trivial.

### 3.1 OPE ALGORITHM AS SUBROUTINE FOR OFFLINE POLICY SELECTION

It is obvious that a sound OPE algorithm can be used for OPS, so the sample complexity of OPS is upper-bounded by the sample complexity of OPE up to a logarithmic factor. We state this formally in the theorem below. The proof can be found in Appendix A.1.

**Theorem 1.** (Upper bound on sample complexity of OPS) Given an MDP $M$, a data distribution $d_b$, and a set of policies $\Pi$, suppose that, for any pair $(\varepsilon, \delta)$, there exists an $(\varepsilon, \delta)$-sound OPE algorithm $\mathcal{L}$ on any OPE instance $I \in \{(M, d_b, \pi) : \pi \in \Pi\}$ with a sample size at most $O(N_{OPE}(S, A, H, \varepsilon, 1/\delta))$. Then there exists an $(\varepsilon, \delta)$-sound OPS algorithm for the OPS problem instance $(M, d_b, \Pi)$ which requires at most $O(N_{OPE}(S, A, H, \varepsilon, |\Pi|/\delta))$ episodes.

In terms of the sample complexity, we have an extra $\sqrt{\log(|\Pi|)/n}$ term for OPS due to the union bound. For hyperparameter selection in practice, the size of the candidate set is often much smaller than $n$, so this extra term is negligible. However, if the set is too large, complexity regularization (Bartlett et al., 2002) may need to be considered.

### 3.2 OFFLINE POLICY SELECTION IS NOT EASIER THAN OPE

We have shown that OPS is sample efficient when OPE is sample efficient. However, it remains unclear whether OPS can be sample efficient when OPE is not. In the following theorem, we lower bound the sample complexity of OPS by the sample complexity of OPE. As a result, both OPS and OPE suffer from the same hardness result, and we cannot expect OPS to be sample efficient under conditions where OPE is not sample efficient.

**Theorem 2** (Lower bound on sample complexity of OPS). Suppose for any data distribution $d_b$ and any pair $(\varepsilon, \delta)$ with $\varepsilon \in (0, V_{max}/2)$ and $\delta \in (0, 1)$, there exists an MDP $M$ and a policy $\pi$ such that any $(\varepsilon, \delta)$-sound OPE algorithm requires at least $\Omega(N_{OPE}(S, A, H, \varepsilon, 1/\delta))$ episodes. Then there exists an MDP $M'$ with $S' = S + 2$, $H' = H + 1$, and a set of candidate policies such that for any pair $(\varepsilon, \delta)$ with $\varepsilon \in (0, V_{max}/3)$ and $\delta \in (0, 1/m)$ where $m := \lceil \log(V_{max}/\varepsilon) \rceil \geq 1$, any $(\varepsilon, \delta)$-sound OPS algorithm also requires at least $\Omega(N_{OPE}(S, A, H, \varepsilon, 1/(m\delta)))$ episodes.

The proof sketch is to construct an OPE algorithm that queries OPS as a subroutine. As a result, the sample complexity of OPS is lower bounded by the sample complexity of OPE. We use the reduction first mentioned in Wang et al. (2020), and present a proof in Appendix A.2.

There exist several hardness results for OPE in tabular settings and with linear function approximation (Yin & Wang, 2021; Wang et al., 2020). Theorem 2 implies that the same hardness results hold for OPS. We should not expect to have a sound OPS algorithm without additional assumptions. Theorem 2, however, does not imply that OPS and OPE are always equally hard. There are instances where OPS is easy but OPE is not. For example, when all policies in the candidate set all have the same value, any random policy selection is sound. However, OPE can still be difficult in such cases.

### 3.3 IMPORTANCE SAMPLING ACHIEVES NEARLY MINIMAX OPTIMAL SAMPLE COMPLEXITY

We present an exponential lower bound on the sample complexity of OPE in Theorem 5 in the appendix, which uses the same construction from Xiao et al. (2022). By the lower bound on the sample complexity of OPE and Theorem 2, we now have a lower bound for OPS.

**Corollary 1** (Exponential lower bound on the sample complexity of OPS). For any positive integers $S, A, H$ with $S > 2H$ and a pair $(\varepsilon, \delta)$ with $0 < \varepsilon \leq \sqrt{1/8}$, $\delta \in (0, 1)$, any $(\varepsilon, \delta)$-sound OPS algorithm needs at least $\tilde{\Omega}(A^{H-1}/\varepsilon^2)$ episodes.

We can use a common OPE method, importance sampling (IS), with a random behavior policy for policy selection. Recall the IS estimator (Rubinstein, 1981) is given by

$$\hat{J}(\pi) = \frac{1}{n} \sum_{i=1}^{n} \underbrace{\prod_{h=0}^{H-1} \frac{\pi(A_h^{(i)}|S_h^{(i)})}{\pi_b(A_h^{(i)}|S_h^{(i)})}}_{W_i} \underbrace{\sum_{h=0}^{H-1} R_h^{(i)}}_{G_i},$$

where $n$ is the number of episodes in the dataset $D$. We now provide an upper bound on the sample complexity of OPS using IS. The proof can be found in Appendix A.3.

**Theorem 3** (OPS using importance sampling). Suppose the data collection policy is uniformly random, that is, $\pi_b(a|s) = 1/A$ for all $(s, a) \in \mathcal{S} \times \mathcal{A}$, and $|G_i| \leq V_{max}$ almost surely. Then the selection algorithm $\mathcal{L}$ that selects the policy with the highest IS estimate is $(\varepsilon, \delta)$-sound with $O(A^H V_{max} \ln(|\Pi|/\delta)/\varepsilon^2)$ episodes.

The theorem suggests that IS achieves a nearly minimax optimal sample complexity for OPS up to a factor $A$ and logarithmic factors. There are other improved variants of IS, including per-decision IS and weighted IS (Precup et al., 2000; Sutton & Barto, 2018). However, none of these variants can help reduce sample complexity in the worst case because the lower bound in Corollary 1 holds for any OPS algorithm. The result suggests that we need to consider additional assumptions on the environment, the data distribution, or the candidate set to obtain guarantees for OPS.

Note that Wang et al. (2017) have shown that IS estimator achievew the minimax mean squared error for the OPE problem. Our result shows that IS also achieves a (nearly) minimax sample complexity for the OPS problem.

There are other examples where OPS is efficient by using particular OPE methods. For example, Liu et al. (2021) consider environments where the state contains exogenous variables; the agent has limited impact on those exogenous variables; and the agent has the knowledge about the endogenous dynamic. In such environments, they provide a sound OPE method to select hyperparameters. Another example is when an accurate simulator of the environment is available. The simulation lemma (Kearns & Singh, 2002) shows that we can evaluate any policy accurately and thus perform sound OPS. All these specialized strategies indicate that we can perform OPS efficiently, under much stronger assumptions.

## 4 CONDITIONS FOR SAMPLE EFFICIENT OFFLINE POLICY SELECTION

In the following sections, we discuss the conditions under which OPS using Fitted Q-iteration and Bellman Error are efficient for offline policy selection. These results are based on existing theoretical analysis of FQE and BE, for example, Chen & Jiang (2019); Le et al. (2019); Xie et al. (2021); Duan et al. (2021).

### 4.1 OFFLINE POLICY SELECTION USING FITTED Q-EVALUATION

Fitted Q-evaluation (FQE) is a commonly-used OPE method that has been shown to be effective for policy selection in benchmark datasets (Paine et al., 2020). FQE applies the empirical Bellman evaluation operator on the value estimate iteratively: given a function class $\mathcal{F}$, we initialize $q_{H-1} = 0$, and, for $h = H - 2, \ldots, 0$, $q_h = \arg\min_{f \in \mathcal{F}} \hat{l}_h(f, q_{h+1})$ where

$$\hat{l}_h(f, q_{h+1}) := \frac{1}{|D_h|} \sum_{(s,a,r,s') \in D_h} (f(s, a) - r - q_{h+1}(s', \pi(s')))^2$$

and $D_h$ is the dataset at horizon $h$ with distribution $\mu_h$. The FQE estimate for $J(\pi)$ is $\hat{J}(\pi) = \mathbb{E}_{a \sim \pi(\cdot|s_0)}[q_0(s_0, a)]$.

Intuitively, we should be able to evaluate the value of all policies covered by the data distribution. If the candidate policy set is a subset of the set of policies covered by data, then we can do policy selection well. To define the notion of well-covered policies, we use the measure of distribution shift introduced in Xie et al. (2021). Given a non-negative real value $C$, we define $\Pi(C, \mu, \mathcal{F})$ be the set of policies covered by the data distribution $\mu$, that is, for any policy $\pi \in \Pi(C, \mu, \mathcal{F})$, $\max_{f,f' \in \mathcal{F}} \frac{\|f - \mathcal{T}^\pi f'\|_{d_h^\pi}}{\|f - \mathcal{T}^\pi f'\|_{\mu_h}} \leq C$ holds for any $h \in [H]$.

The theorem below provides a theoretical guarantee for OPS using FQE. The proof can be found in Appendix A.4.

**Theorem 4** (Offline policy selection for well-covered policies). Suppose we have a function class $\mathcal{F}$ with Rademacher complexity $R_n^\mu(\mathcal{F})$, an approximation error $\varepsilon_{apx}$, and an optimization error $\varepsilon_{opt}$ (see Appendix A.4 for the definitions). If there exists a non-negative value $C$ such that $\Pi \subset$

$\Pi(C, \mu, \mathcal{F})$, then the OPS algorithm $\mathcal{L}$ that selects the policy with the highest FQE estimate satisfies

$$\Pr\left(J(\mathcal{L}(D, \Pi)) \geq J(\pi^{\dagger}) - 2H\sqrt{C}\sqrt{\varepsilon_{opt} + \varepsilon_{apx} + c_0 H R_n^{\mu}(\mathcal{F}) + c_0 H^2 \sqrt{\frac{\log(|\Pi|H/\delta)}{n}}}\right) \geq 1 - \delta$$

for some constant $c_0$.

If we have a good function class and an optimizer, then FQE only requires a polynomial number of episodes for OPS under the assumption that all candidate policies are well-covered by the data. However, in practice, we need to choose the function class and optimization hyperparameters. It is easy to choose optimization hyperparameters such as the learning rate. We can do so by choosing the optimizer with the lowest $\varepsilon_{opt}$. This can be done by selecting the optimization hyperparameters that result in the smallest TD errors $\sum_h \hat{l}_h(q_h, q_{h+1})$ on a validation dataset. However, selecting a function class is nontrivial, because we need to balance the approximation error, the complexity measure and how well the data covers the candidate policies. In our experiments, we fix the function class as a neural network model with the same architecture and tune the optimization hyperparameters.

Theorem 4 presents a positive result that OPS can be sample efficient when selecting amongst well-covered policies. However, if one of the candidate policies is not well-covered, then the FQE estimate may overestimate the value of the uncovered policy and resulting in poor OPS. It is known that FQE can even diverge (Sutton & Barto, 2018), due to the fact that it combines off-policy learning with bootstrapping and function approximation, known as the deadly triad. One way to circumvent the issue of uncovered policies is to find a way to assign low values for uncovered policies. One heuristic way is to early stop when FQE diverges and assign lower value estimates to policies for which FQE diverges since the policy is likely to be not covered by the data. Another heuristic is to use a pessimistic version of FQE that penalizes policies whose data distribution is dissimilar from the behavior distribution. This approach is similar to CQL, but it introduces an additional hyperparameter to control the level of pessimism, which again is not easy to tune in the offline setting.

**Practical insight.** Our result explains the empirical findings in Paine et al. (2020) that FQE is sufficient for selecting hyperparameters for conservative algorithms and imitation learning algorithms. A recent paper (Kumar et al., 2022) discusses when we should use pessimistic offline RL or behavioral cloning under specific conditions. Our result shows that we can in fact run both algorithms and choose the one with highest FQE estimate. With high probability, we can choose the best policy among offline RL and behavioral cloning algorithms without considering specific conditions.

## 4.2 OFFLINE POLICY SELECTION USING THE BELLMAN ERROR

In this section, we investigate whether OPS using the estimated Bellman error has any advantage over FQE. Most value-based methods output value functions and we often perform policy selection over value functions by considering the greedy policies with respect to these value functions. Suppose we are given a candidate set of value functions $\{q_i\}_{i=1}^K$, a natural criterion is to select the value function that is closest to the optimal value function, that is, $\|q_i - q^*\|_{\infty}$ is the smallest. A justification of choosing the smallest $\|q_i - q^*\|_{\infty}$ is that the value error gives us a lower bound on $J(\pi_i)$ where $\pi_i$ is the greedy policy with respect to $q_i$. That is, $J(\pi_i) \geq J(\pi^*) - 2H\|q_i - q^*\|_{\infty}$ (Singh & Yee, 1994). Therefore, choosing the smallest $\|q_i - q^*\|_{\infty}$ is equivalent to choosing the largest lower bound on the policy value. While we do not know $q^*$, it might still be possible to derive a similar lower bound on the policy value, possibly by estimating the Bellman error (BE).

A potential use case of the BE is that if the data distribution covers the entire state-action space well, then the BE is a good surrogate for the policy value. More formally, the concentration coefficient $C$ is defined as the smallest value such that $\max_{s,a,h} \frac{d_h^{\pi}(s,a)}{\mu_h(s,a)} \leq C$ for all policies $\pi$. By the performance difference lemma (Kakade, 2003), we can show that

$$J(\pi) \geq J(\pi^*) - CH \sum_{h=0}^{H-1} \|q_h - \mathcal{T}q_{h+1}\|_{2,\mu_h}.$$

That is, the Bellman error evaluated on the dataset can be a useful indicator of the quality of the greedy policy, if $C$ is small. Note that a small concentration coefficient is a stronger assumption compared to the assumption of FQE. To be more specific, the condition $\max_{s,a,h} \frac{d_h^{\pi}(s,a)}{\mu_h(s,a)} \leq C$ for

all policies $\pi \in \Pi$ implies $\max_{f,f' \in \mathcal{F}} \frac{\|f - \mathcal{T}^\pi f'\|_{d_h^\pi}}{\|f - \mathcal{T}^\pi f'\|_{\mu_h}} \leq C$ for all policies $\pi \in \Pi$. That is, we only need coverage for all policies in the candidate set for sample efficient OPS with FQE. However, for BE, the measure of distribution shift needs to be bounded for all policies.

Finding the value function with the lowest Bellman error in deterministic environments is sample efficient. To see why this is the case, let $k$ be the output index of the selection algorithm that selects the index of the value function with the smallest empirical Bellman error, then with probability at least $1 - \delta$, for some constant $c_0$,

$$\|q_k - \mathcal{T} q_k\|_{2,\mu}^2 \leq \min_{i=1,\ldots,|\Pi|} \|q_i - \mathcal{T} q_i\|_{2,\mu}^2 + c_0 V_{max} \sqrt{\frac{\log(|\Pi| H / \delta)}{n}}$$

where we denote $\|q_i - \mathcal{T} q_i\|_{2,\mu}^2 = \frac{1}{H} \sum_h \|q_{i,h} - \mathcal{T} q_{i,h+1}\|_{2,\mu_h}^2$ for simplicity. Therefore, the sample size $n$ only needs to scale with $V_{max}^2$. In contrast, the sample size of FQE depends on the complexity of the function class $R_n^\mu(\mathcal{F})$. If we use a function class with high complexity to perform FQE, then we expect to see that BE is a more sample efficient method for selection compared to FQE when the data covers the state-action space well, that is, when $C$ is small.

If the concentration coefficient is large or infinite, then the Bellman error can be a poor indicator of the greedy policy's performance. For example, Fujimoto et al. (2022) provide several examples where BE is poor surrogate for value error even when the environment is deterministic. Moreover, several empirical works have shown that OPS using the Bellman error or value estimate from offline RL algorithms underperform using OPE estimates (Tang & Wiens, 2021). Our result provides theoretical reasoning for these experimental observations. Since, in most practical scenarios the concentration coefficient can be very large or infinite, OPS using the BE can be poor.

**Estimating Bellman error in stochastic environments.** Estimating Bellman error in stochastic environments potentially requires more samples. This is because doing so often involves fitting a function (Antos et al., 2008; Farahmand & Szepesvári, 2011), and resultingly the sample complexity of estimating the BE accurately depends on the complexity of the function class. We hypothesize that it makes the sample complexity similar to FQE. If the sample complexity is similar to FQE and policy selection using BE requires stronger assumptions, then it is natural to use FQE in stochastic environments even if the data coverage is good. We leave the exact analysis of the sample complexity of OPS using estimated Bellman Error in stochastic environments for future work.

## 5 EMPIRICAL COMPARISON BETWEEN FQE AND BE

In this section, we aim to validate our results experimentally by answering the following questions: (1) How do FQE and BE perform when (a) all candidate policies are well-covered, (b) the data contains diverse trajectories or (c) the data neither covers all candidate policies nor provides diverse trajectories? (2) Can BE be more sample efficient than FQE for OPS?

We conduct experiments on two standard RL environments: Acrobot and Cartpole. We first generate a set of candidate policy-value pairs $\{\pi_i, q_i\}_{i=1}^K$ by running CQL (Kumar et al., 2020) with different hyperparameters on a batch of data collected by an optimal policy with random actions taken $40\%$ of the time. We then use either FQE or BE to rank the candidate policies, and select the top-$k$ policies. The OPS problem we described in this paper is a special case with $k = 1$. We also included a random selection baseline where the we randomly choose $k$ policies from the candidate set. To evaluate the performance of top-$k$ policy selection, we consider the top-$k$ regret which is used in the existing literature (Zhang & Jiang, 2021; Paine et al., 2020). Top-$k$ regret is the gap between the best policy within the top-k policies, and the best among all candidate policies. More experimental details can be found in Appendix B.

In the first set of experiments, we generate three different datasets: (a) well-covered data is generated such that all candidate policies are well-covered, (b) diverse data includes more diverse trajectories to obtain a lower concentration coefficient, and (c) expert data is collected by a deterministic and optimal policy. Figure 1 shows the top-$k$ regret with 900 episodes of data. FQE performs very well with a small regret on well-covered and diverse data. BE does not perform well with well-covered data. However, with diverse data, BE performs much better. Surprisingly, for expert data, BE performs better than FQE even though the data distribution has an extremely large concentration

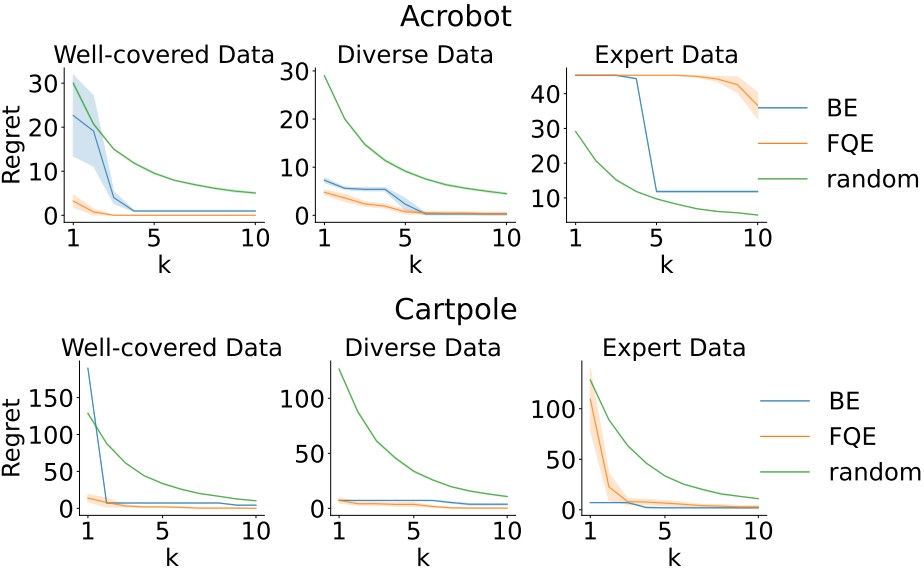

Figure 1: Top-$k$ regret with varying $k$ and $900$ episodes on Acrobot and Cartpole. The results are averaged over 5 runs with one standard error.

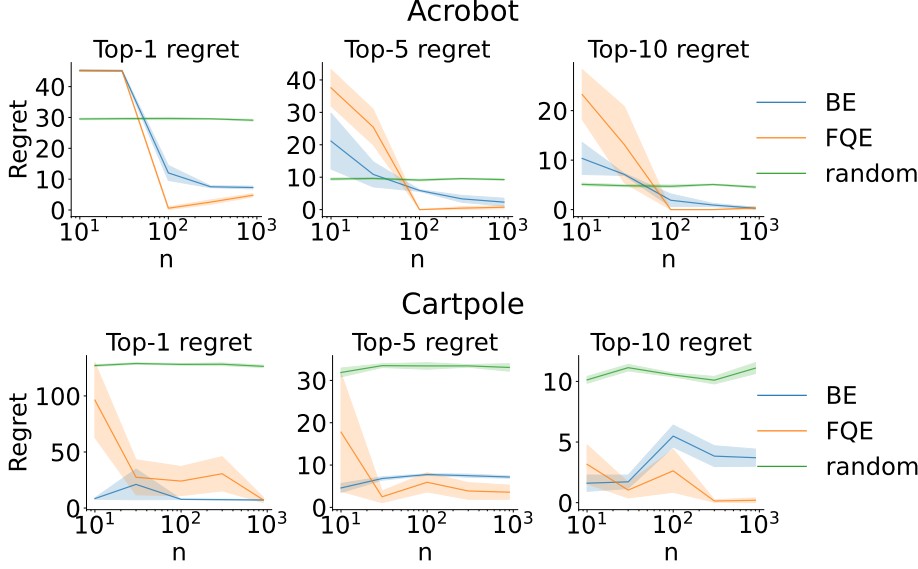

Figure 2: Top-$k$ regret with varying sample size on Acrobot and Cartpole with diverse data.

coefficient. In Acrobot, BE tends to perform better with a larger $k$ even with good data coverage, while FQE can have a small top-1 regret as long as the data covers the candidate policy.

In the second set of experiments, we use diverse data. Figure 2 shows the results with varying numbers of episodes for top-$k$ regret with $k \in \{1, 5, 10\}$. BE performs better than FQE when the sample size is small (less than 100 episodes) in most cases. The random selection baseline is expected to perform independent of the data used for OPS. We expect to see a larger gap when the environment has a high-dimensional state spacee so a function class with large complexity is needed for FQE.

In summary, these experimental results validate our theoretical results. We show (1) FQE performs very well when the candidate policies are well-covered; (2) BE performs well under a diverse data coverage assumption; and (3) neither perform well without their corresponding data coverage conditions. Moreover, BE can be more sample efficient than FQE under good data coverage.

## 6 RELATED WORK

In this section we provide a more comprehensive survey of prior work on model selection for reinforcement learning. In the online setting, model selection has been studied extensively across contextual bandits (Foster et al., 2019) to reinforcement learning (Lee et al., 2021). In the online setting, the goal is to select model classes while balancing exploration and exploitation to achieve low regret, which is very different from the offline setting where no exploration is performed.

In the offline setting, besides using OPE for model selection, Farahmand & Szepesvári (2011) and Zhang & Jiang (2021) consider selecting a value function that has the smallest Bellman error or is the closest to the optimal value function. Farahmand & Szepesvári (2011) consider selecting a value function among a set of candidate value functions in stochastic environments such that with high probability, $\|q_k - \mathcal{T}q_k\|_{2,\mu}^2 \le \min_i \|q_i - \mathcal{T}q_i\|_{2,\mu}^2 + \varepsilon$ where $k$ is the output of a selection algorithm. To estimate the Bellman error, they propose to fit a regression model $\tilde{q}_i$ to predict $\mathcal{T}q_i$ and bound the Bellman error by $\|q_i - \tilde{q}_i\|_{2,\mu} + b_i$ where $b_i$ is an upper bound on the estimation error of the regression. However, as we show in this paper, even in a deterministic environment where selecting the smallest Bellman error is easy, it does not imply effective OPS unless the data coverage is good.

Zhang & Jiang (2021) consider the projected Bellman error with a piecewise constant function class. They show that if $q^*$ is in the candidate set and a stronger data assumption is satisfied, then they can choose the optimal value function by selecting the value function with the smallest Batch Value-Function Tournament loss (BVFT). Xie & Jiang (2021) consider BVFT with approximation error. In practice, their method is computationally expensive since it scales with $O(|\Pi|^2)$ instead of $O(|\Pi|)$, making the method impractical when the candidate set is not small.

Other work on model selection in RL is in other settings: selecting models and selecting amongst OPE estimators. Hallak et al. (2013) consider model selection for model-based RL algorithms with batch data. They focus on selecting the most suitable model that generates the observed data, based on the maximum likelihood framework. Su et al. (2020) consider estimator selection for OPE when the estimators can be ordered with monotonically increasing biases and decreasing confidence intervals. They also show that their method achieves an oracle inequality for estimator selection.

To the best of our knowledge, there is no previous work on understanding the fundamental limits for the OPS problem in RL. There is one related work in the batch contextual bandit setting, studying the selection of a linear model (Lee et al., 2022). They provide a hardness result suggesting it is impossible to achieve an oracle inequality that balances the approximation error, the complexity of the function class, and data coverage. However, their results are restricted only to the contextual bandit setting. In this paper, we consider the more general problem, selecting a policy from a set of policies, in the RL setting.

## 7 CONCLUSION & DISCUSSION

In this paper, we have made progress towards understanding when OPS is feasible for RL. Our main result that the sample complexity of OPS is lower-bounded by the sample complexity of OPE, is perhaps expected. However, to our knowledge, this has never been formally shown. This result implies that without conditions to make OPE feasible, we cannot do policy selection efficiently. We provided theoretical statements with experimental evidence demonstrating that FQE is sample efficient when all candidate policies are well-covered and that BE can be more sample efficient under stronger data coverage.

A natural next step from this work is to consider the structure or properties of the candidate policy set. For example, Kumar et al. (2021) use the trend in the TD error to select the number of training steps for CQL. The early stopping criterion could be selected based on errors on the dataset, and it is possible that such an approach could be proved to be sound in certain settings. As another example, if we know that performance is smooth in a hyperparameter for an algorithm, such as a regularization parameter, then it might be feasible to exploit curve fitting to get better estimates of performance. Our work does not yet answer these questions, but it does provide some insights into OPS that could lead to the development of sound hyperparameter selection procedures for real-world RL applications.

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

# A TECHNICAL DETAILS

## A.1 PROOF OF THEOREM 1

**Theorem 1.** (Upper bound on sample complexity of OPS) Given an MDP $M$, a data distribution $d_b$, and a set of policies $\Pi$, suppose that, for any pair $(\varepsilon, \delta)$, there exists an $(\varepsilon, \delta)$-sound OPE algorithm $\mathcal{L}$ on any OPE instance $I \in \{(M, d_b, \pi) : \pi \in \Pi\}$ with a sample size at most $O(N_{OPE}(S, A, H, \varepsilon, 1/\delta))$. Then there exists an $(\varepsilon, \delta)$-sound OPS algorithm for the OPS problem instance $(M, d_b, \Pi)$ which requires at most $O(N_{OPE}(S, A, H, \varepsilon, |\Pi|/\delta))$ episodes.

*Proof.* The OPS algorithm $\mathcal{L}(D, \Pi)$ for a given $(\varepsilon, \delta)$ works as follows: we query an $(\varepsilon', \delta')$-sound OPE algorithm for each policy in $\Pi$ and select the policy with the highest estimated value. That is, $\mathcal{L}(D, \Pi)$ outputs the policy $\bar{\pi} := \arg\max_{\pi \in \Pi} \hat{J}(\pi)$, where $\hat{J}(\pi)$ is the value estimate for policy $\pi$ by the $(\varepsilon', \delta')$-sound OPE algorithm with data $D$.

By definition of an $(\varepsilon', \delta')$-sound OPE algorithm we have

$$\Pr_{D \sim d_b}(|\hat{J}(\pi) - J(\pi)| \leq \varepsilon') \geq 1 - \delta', \forall \pi \in \Pi.$$

Applying the union bound, we have

$$\Pr_{D \sim d_b}(\forall \pi \in \Pi, |\hat{J}(\pi) - J(\pi)| \leq \varepsilon') \geq 1 - \delta'|\Pi|. \tag{1}$$

Let $\pi^\dagger$ denote the best policy in the candidate set $\Pi$, that is, $\pi^\dagger := \arg\max_{\pi \in \Pi} J(\pi)$. With probability $1 - \delta'|\Pi|$, we have

$$J(\bar{\pi}) \geq \hat{J}(\bar{\pi}) - \epsilon' \geq \hat{J}(\pi^\dagger) - \varepsilon' \geq J(\pi^\dagger) - 2\varepsilon'$$

The first and third inequalities follow from equation 1, and the second inequality follows from the definition of $\bar{\pi}$.

Finally, by setting $\delta' = \delta/|\Pi|$ and $\varepsilon' = \varepsilon/2$, we get

$$\Pr_{D \sim d_b}(J(\bar{\pi}) \geq J(\pi^\dagger) - \varepsilon) \geq 1 - \delta.$$

That is, $\mathcal{L}$ is an $(\varepsilon, \delta)$-sound OPS algorithm. This requires at most $O(N_{OPE}(S, A, H, \varepsilon/2, |\Pi|/\delta))$ samples. $\qquad \square$

## A.2 PROOF OF THEOREM 2

**Theorem 2** (Lower bound on sample complexity of OPS). Suppose for any data distribution $d_b$ and any pair $(\varepsilon, \delta)$ with $\varepsilon \in (0, V_{max}/2)$ and $\delta \in (0, 1)$, there exists an MDP $M$ and a policy $\pi$ such that any $(\varepsilon, \delta)$-sound OPE algorithm requires at least $\Omega(N_{OPE}(S, A, H, \varepsilon, 1/\delta))$ episodes. Then there exists an MDP $M'$ with $S' = S + 2$, $H' = H + 1$, and a set of candidate policies such that for any pair $(\varepsilon, \delta)$ with $\varepsilon \in (0, V_{max}/3)$ and $\delta \in (0, 1/m)$ where $m := \lceil \log(V_{max}/\varepsilon) \rceil \geq 1$, any $(\varepsilon, \delta)$-sound OPS algorithm also requires at least $\Omega(N_{OPE}(S, A, H, \varepsilon, 1/(m\delta)))$ episodes.

*Proof.* Our goal is to construct an $(\varepsilon, \delta)$-sound OPE algorithm with $\delta \in (0, 1)$ and $\varepsilon \in [0, V_{max}/2]$. To evaluate any policy $\pi$ in $M$ with dataset $D$ sampled from $d_b$, we first construct a new MDP $M_r$ with two additional states: an initial state $s_0$ and a terminal state $s_1$. Taking $a_1$ at $s_0$ transitions to $s_1$ with reward $r$. Taking $a_2$ at $s_0$ transitions to the initial state in the *original* MDP $M$.

Let $\Pi = \{\pi_1, \pi_2\}$ be the candidate set in $M_r$ where $\pi_1(s_0) = a_1$ and $\pi_2(s_0) = a_2$ and $\pi_2(a|s) = \pi(a|s)$ for all $(s, a) \in \mathcal{S} \times \mathcal{A}$. Since $\pi_1$ always transitions to $s_1$, it never transitions to states in MDP $M$. Therefore, $\pi_1$ can be arbitrary for all $(s, a) \in \mathcal{S} \times \mathcal{A}$. We can add any number of transitions $(s_0, a_1, r, s)$ and $(s_0, a_2, 0, s)$ in $D$ to construct the dataset $D_r$ with distribution $d_{b,r}$ arbitrarily.

Suppose we have an $(\varepsilon', \delta')$-sound OPS algorithm, where we set $\varepsilon' = 2\varepsilon/3$, $\delta' = \delta/m$ and $m := \lceil \log(V_{max}/\varepsilon') \rceil$. Note that if this assumption does not hold, then it directly implies that the sample complexity of OPS is larger than $\Omega(N_{OPE}(S, A, H, \varepsilon, 1/\delta))$. Our strategy will be to iteratively set

the reward $r$ of MDP $M_r$ and run our sound OPS algorithm on $\Pi$ and using bisection search to estimate a precise interval for $J(\pi)$.

The process is as follows. By construction, our OPS algorithm will output either $\pi_1$, which has value $J_{M_r}(\pi_1) = r$, or output $\pi_2$, which has value $J_{M_r}(\pi_2) = J_M(\pi)$. That is, it has the same value as $\pi$ in the original MDP. Let us consider the following two cases. Let $\pi^\dagger$ be the best policy in $\Pi$ for MDP $M_r$.

**Case 1: the OPS algorithm selects $\pi_1$.** We know, by definition of a sound OPS algorithm, that

$$\Pr(J_{M_r}(\pi_1) \geq J_{M_r}(\pi^\dagger) - \varepsilon') \geq 1 - \delta'$$
$$\implies \Pr(r \geq \max(r, J_{M_r}(\pi_2)) - \varepsilon') \geq 1 - \delta'$$
$$\implies \Pr(J_{M_r}(\pi_2) \leq r + \varepsilon') \geq 1 - \delta'.$$

**Case 2: the OPS algorithm selects $\pi_2$.**

$$\Pr(J_{M_r}(\pi_2) \geq J_{M_r}(\pi^\dagger) - \varepsilon') \geq 1 - \delta'$$
$$\implies \Pr(J_{M_r}(\pi_2) \geq \max(r, J_{M_r}(\pi_2)) - \varepsilon') \geq 1 - \delta'$$
$$\implies \Pr(J_{M_r}(\pi_2) \geq r - \varepsilon') \geq 1 - \delta'.$$

Given this information, we describe the iterative process by which we produce the estimate $\hat{J}(\pi)$. We first set $U = V_{max}, L = 0$ and $r = \frac{U+L}{2}$ and run the sound OPS algorithm with input $D_r$ of sample size $n_r$ and the candidate set $\Pi$. Then if the selected policy is $\pi_1$, then we conclude the desired event $J(\pi) \leq r + \varepsilon'$ occurs with probability at least $1 - \delta'$, and set $U$ equal to $r$. If the selected policy is $\pi_2$, then we know the desired event $J(\pi) \geq r - \varepsilon'$ occurs with probability at least $1 - \delta'$, and set $L$ equal to $r$. We can continue the bisection search until the accuracy is less than $\varepsilon'$, that is, $U - L \leq \varepsilon'$, and the output value estimate is $\hat{J}(\pi) = \frac{U+L}{2}$.

If all desired events at each call occur, then we conclude that $L - \varepsilon' \leq J(\pi) \leq U + \varepsilon'$ and thus $|J(\pi) - \hat{J}(\pi)| \leq \varepsilon$. The total number of OPS calls is at most $m$. Setting $\delta' = \delta/m$ and applying a union bound, we can conclude that with probability at least $1 - \delta$, $|J(\pi) - \hat{J}(\pi)| \leq \varepsilon$.

Finally, since any $(\varepsilon, \delta)$-sound OPE algorithm on the instance $(M, d_b, \pi)$ needs at least $\Omega(N_{OPE}(S, A, H, \varepsilon, 1/\delta))$ samples, the $(\varepsilon', \delta')$-sound OPS algorithm needs at least $\Omega(N_{OPE}(S, A, H, \varepsilon, 1/\delta))$, or $\Omega(N_{OPE}(S, A, H, 3\varepsilon/2, 1/m\delta'))$ samples for at least one of the instances $(M_r, d_{b,r}, \Pi)$. □

### A.3 PROOF OF THEOREM 3

**Theorem 3** (OPS using importance sampling). Suppose the data collection policy is uniformly random, that is, $\pi_b(a|s) = 1/A$ for all $(s, a) \in \mathcal{S} \times \mathcal{A}$, and $|G_i| \leq V_{max}$ almost surely. Then the selection algorithm $\mathcal{L}$ that selects the policy with the highest IS estimate is $(\varepsilon, \delta)$-sound with $O(A^H V_{max} \ln(|\Pi|/\delta)/\varepsilon^2)$ episodes.

*Proof.* Since the policy is uniform random, we know $|W_i G_i| < A^H V_{max}$ almost surely. Moreover, the importance sampling estimator is unbiased, that is, $\mathbb{E}[W_i G_i] = J(\pi)$. Using the Bernstein's inequality, we can show that the IS estimator satisfies

$$\Pr\left(|\hat{J}(\pi_k) - J(\pi_k)| \leq \frac{2A^H V_{max} \ln(2/\delta)}{3n} + \sqrt{\frac{2\mathrm{Var}(W_i G_i) \ln(2/\delta)}{n}}\right) \geq 1 - \delta$$

for one candidate policy $\pi_k$. Using the union bound, we have

$$\Pr\left(|\hat{J}(\pi_k) - J(\pi_k)| \leq \frac{2A^H V_{max} \ln(2|\Pi|/\delta)}{3n} + \sqrt{\frac{2\mathbb{V}(W_i G_i) \ln(2|\Pi|/\delta)}{n}}, \forall k\right) \geq 1 - \delta$$

That is,

$$\Pr\left(J(\mathcal{L}(D, \Pi)) \geq J(\pi^\dagger) - \frac{4A^H V_{max} \ln(2|\Pi|/\delta)}{3n} + \sqrt{\frac{8\mathbb{V}(W_i G_i) \ln(2|\Pi|/\delta)}{n}}\right) \geq 1 - \delta.$$

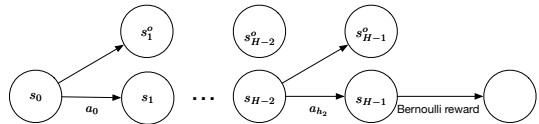

Figure 3: Lower bound construction.

For the variance term,

$$\mathbb{V}(W_i G_i) = \mathbb{E}[W_i^2 G_i^2] - \mathbb{E}[W_i G_i]^2 \leq \mathbb{E}[W_i^2 G_i^2] \leq \sqrt{\mathbb{E}[W_i^2]\mathbb{E}[G_i^2]} \leq A^H V_{max}$$

The second inequality follows from the Cauchy-Schwarz inequality. Therefore, if $n > 32 A^H V_{max} \ln(2|\Pi|/\delta)/\varepsilon^2$, $\mathcal{L}$ is $(\varepsilon, \delta)$-sound. $\square$

**Theorem 5** (Exponential lower bound on the sample complexity of OPE). For any positive integers $S, A, H$ with $S > 2H$ and a pair $(\varepsilon, \delta)$ with $0 < \varepsilon \leq \sqrt{1/8}$, $\delta \in (0, 1)$, any $(\varepsilon, \delta)$-sound OPE algorithm needs at least $\Omega(A^H \ln(1/\delta)/\varepsilon^2)$ episodes.

*Proof.* We provide a proof which uses the construction from Xiao et al. (2022). They provide the result for the offline RL problem with Gaussian rewards. Here we provide the result for OPE problem with Bernoulli rewards since we assume rewards are bounded to match Theorem 3.

We can construct an MDP with $S$ states, $A$ actions and $2H$ states in Figure 3. Given any behavior policy $\pi_b$, let $a_h = \arg\min_a \pi_b(a|s_h)$ be the action that leads to the next state $s_{h+1}$ from state $s_h$, and all other actions lead to an absorbing state $s_h^o$. Once we reach an absorbing state, the agent gets zero reward for all actions for the remainder of the episode. The only nonzero reward is in the last state $s_{H-1}$. Consider a target policy that chooses $a_h$ for state $s_h$ for all $h = 0, \ldots, H-1$, and two MDPs where the only difference between them is the reward distribution in $s_{H-1}$. MDP 1 has Bernoulli distribution with mean $1/2$ and MDP 2 has Bernoulli distribution with mean $1/2 - 2\varepsilon$. Let $\mathbb{P}_1$ denote the probability measure with respect to MDP 1 and $\mathbb{P}_2$ denote the probability measure with respect to MDP 2.

Let $\hat{r}$ denote the OPE estimate by an algorithm $\mathcal{L}$. Define an event $E = \{\hat{r} < \frac{1}{2} - \varepsilon\}$. Then $\mathcal{L}$ is not $(\varepsilon, \delta)$-sound if

$$\frac{\mathbb{P}_1(E) + \mathbb{P}_2(E^c)}{2} \geq \delta.$$

This is because $\mathcal{L}$ is not $(\varepsilon, \delta)$-sound if either $\mathbb{P}_1(\hat{r} < \frac{1}{2} - \varepsilon) \geq \delta$ or $\mathbb{P}_2(\hat{r} > \frac{1}{2} - \varepsilon) \geq \delta$.

Using the Bretagnolle–Huber inequality (See Theorem 14.2 of Lattimore & Szepesvári (2020)), we know

$$\frac{\mathbb{P}_1(E) + \mathbb{P}_2(E^c)}{2} \geq \frac{1}{4} \exp(-D_{KL}(\mathbb{P}_1, \mathbb{P}_2)).$$

By the chain rule for KL-divergence and the fact that $\mathbb{P}_1$ and $\mathbb{P}_2$ only differ in the reward for $(s_{H-1}, a_{H-1})$, we have

$$D_{KL}(\mathbb{P}_1, \mathbb{P}_2) = \mathbb{E}_1\left[\sum_{i=1}^{n} \mathbb{I}\{S_{H-1}^{(i)} = s_{H-1}, A_{H-1}^{(i)} = a_{H-1}\}\left(\frac{1}{2}\log\left(\frac{1/2}{1/2-\varepsilon}\right) + \frac{1}{2}\log\left(\frac{1/2}{1/2+\varepsilon}\right)\right)\right]$$

$$= \sum_{i=1}^{n} \mathbb{P}_1(S_{H-1}^{(i)} = s_{H-1}, A_{H-1}^{(i)} = a_{H-1})\left(-\frac{1}{2}\log(1 - 4\varepsilon^2)\right)$$

$$\leq \frac{n 8 \varepsilon^2}{A^H}$$

The last inequality follows from $-\log(1 - 4\varepsilon^2) \leq 8\varepsilon^2$ if $4\varepsilon^2 \leq 1/2$ (Krishnamurthy et al., 2016) and $\mathbb{P}_1(S_{H-1}^{(i)} = s_{H-1}, A_{H-1}^{(i)} = a_{H-1}) < 1/A^H$ from the construction of the MDPs.

Finally,

$$\frac{\mathbb{P}_1(E) + \mathbb{P}_2(E^c)}{2} \geq \frac{1}{4} \exp\left(-\frac{n8\varepsilon^2}{A^H}\right)$$

which is larger than $\delta$ if $n \leq A^H \ln(1/4\delta)/8\varepsilon^2$. As a result, we need at least $\Omega(A^H \ln(1/\delta)/\varepsilon^2)$ episodes. $\qquad\square$

## A.4 Proof of Theorem 4

**Assumption 1** (Approximation error)**.** For any policy $\pi \in \Pi$ and $h \in [H]$, we assume the approximation error is bounded by $\varepsilon_{apx}$, that is,

$$\sup_{g\in\mathcal{F}} \inf_{f\in\mathcal{F}} \|f - \mathcal{T}^\pi g\|_{2,\mu_h}^2 \leq \varepsilon_{apx}.$$

**Assumption 2** (Optimization error)**.** Given a target value function $g$, we want to find the empirical risk minimizer

$$\tilde{f} = \arg\min_{f\in\mathcal{F}} \hat{l}(f,g).$$

We assume we have an optimization oracel that can return a value function $\hat{f}$ such that the optimization error is bounded, that is,

$$\hat{l}(\hat{f},g) \leq \hat{l}(\tilde{f},g) + \varepsilon_{opt}.$$

**Definition 1** (Rademacher complexity)**.** Given a function class $\mathcal{F}$, let $X = \{x_1, \ldots, x_n\}$ denotes $n$ fixed data points at horizon $h$ following the distribution $\mu_h$, the empirical Rademacher complexity is defined as

$$R_X(\mathcal{F}) = \mathbb{E}\left[\sup_{f\in\mathcal{F}} \frac{1}{n} \sum_{i=1}^{n} \sigma_i f(x_i) \mid X\right]$$

where the expectation is with respect to the Rademacher random variables $\sigma_i$. The Rademacher complexity is defined as $R_n^{\mu_h}(\mathcal{F}) = \mathbb{E}[R_X(\mathcal{F})]$ where the expectation is with respect to the $n$ data points. Finally, to simply the notation, we define $R_n^\mu(\mathcal{F}) = \max_{h\in[H]} R_n^{\mu_h}(\mathcal{F})$ as the maximum Rademacher complexity over all horizons.

**Lemma 1** (Excess risk bound, modification from Theorem 5.2 of Duan et al. (2021))**.** Let $q = (q_0, \ldots, q_{H-1})$ be the output of FQE with $n$ sample drawn from the data distribution $\mu_h$ at each horizon $h$. Then we have, with probability $1 - \delta$, the following holds for all $h = 0, \ldots, H-1$

$$\|q_h - \mathcal{T}^\pi q_{h+1}\|_{2,\mu_h}^2 \leq \varepsilon_{opt} + \varepsilon_{apx} + c_0 H R_n^{\mu_h}(\mathcal{F}) + c_0 H^2 \sqrt{\frac{\log(H/\delta)}{n}}$$

for some constant $c_o > 0$.

*Proof.* For each horizon $h \in [H]$, let $q_h$ be the output value function such that $\hat{l}(q_h, q_{h+1}) \leq \hat{l}(\hat{q}_h, q_{h+1}) + \varepsilon_{opt}$ where $\hat{q}_h$ is the empirical minimizer, that is, $\hat{q}_h = \arg\min_{f\in\mathcal{F}} l_h(f, q_{h+1})$. It follows $\hat{l}(q_h, \hat{f}_{h+1}) \leq \hat{l}(q_h^\dagger, q_{h+1}) + \varepsilon_{opt}$ where $q_h^\dagger$ is the population minimizer, that is, $q_h^\dagger = \arg\min_{f\in\mathcal{F}} \|f - \mathcal{T}q_{h+1}\|_{2,\mu_h}$. Then follow the proof of Theorem 5.2 in Duan et al. (2021) by takeing $f_h = q_h$, we have

$$\|q_h - \mathcal{T}^\pi q_{h+1}\|_{2,\mu_h}^2 \leq \varepsilon_{opt} + \varepsilon_{apx} + c H R_n^{\mu_h}(\mathcal{F}) + c H^2 \sqrt{\frac{\log(H/\delta)}{n}}$$

holds for all $h = 0, \ldots, H-1$ with probability at least $1 - \delta$. $\qquad\square$

**Theorem 4** (Offline policy selection for well-covered policies)**.** Suppose we have a function class $\mathcal{F}$ with Rademacher complexity $R_n^\mu(\mathcal{F})$, an approximation error $\varepsilon_{apx}$, and an optimization error $\varepsilon_{opt}$ (see Appendix A.4 for the definitions). If there exists a non-negative value $C$ such that $\Pi \subset \Pi(C, \mu, \mathcal{F})$, then the OPS algorithm $\mathcal{L}$ that selects the policy with the highest FQE estimate satisfies

$$\Pr\left(J(\mathcal{L}(D, \Pi)) \geq J(\pi^\dagger) - 2H\sqrt{C}\sqrt{\varepsilon_{opt} + \varepsilon_{apx} + c_0 H R_n^\mu(\mathcal{F}) + c_0 H^2 \sqrt{\frac{\log(|\Pi|H/\delta)}{n}}}\right) \geq 1 - \delta$$

for some constant $c_0$.

*Proof.* First fix a policy $\pi \in \Pi$. Let $q = (q_0, \ldots, q_{H-1})$ be the output of FQE with $n$ sample drawn from the data distribution $\mu_h$ at each horizon $h$, then $\|q_h - \mathcal{T}^\pi q_{h+1}\|_{d_b}^2 \leq \varepsilon_{opt} + \varepsilon_{apx} + c_0 H R_n^{\mu_h}(\mathcal{F}) + c_0 H^2 \sqrt{\frac{\log(H/\delta)}{n}}$ from Lemma 1 for some constant $c_0$.

By the definition of $\Pi(C, \mu, \mathcal{F})$, we know $\|q_h - \mathcal{T}^\pi q_{h+1}\|_{d_h^\pi} \leq C\|q_h - \mathcal{T}^\pi q_{h+1}\|_{\mu_h}$ for any $f \in \mathcal{F}$. Therefore, we have $\|q_h - \mathcal{T}^\pi q_{h+1}\|_{d_h^\pi}^2 \leq C(\varepsilon_{opt} + \varepsilon_{apx} + c_0 H R_n^{\mu_h}(\mathcal{F}) + c_0 H^2 \sqrt{\frac{\log(H/\delta)}{n}})$.

We know
$$\|q_0^\pi - q_0\|_{1,d_0^\pi} = \sum_a \pi(a|s_0)|q_0^\pi(s_0, a) - q_0(s_0, a)|$$
$$= \sum_a \pi(a|s_0)|(\mathcal{T}^\pi q_1^\pi)(s_0, a) - (\mathcal{T}^\pi q_1)(s_0, a) + (\mathcal{T}^\pi q_1)(s_0, a) - q_0(s_0, a)|$$
$$\leq \sum_{a,s',a'} \pi(a|s_0)p(s'|s,a)\pi(a'|s')|q_1^\pi(s, a) - q_1(s, a)| + \sum_a \pi(a|s_0)|(\mathcal{T}^\pi q_1)(s_0, a) - q_0(s_0, a)|$$
$$= \|q_1^\pi - q_1\|_{1,d_1^\pi} + \|\mathcal{T}^\pi q_1 - q_0\|_{1,d_0^\pi}$$

Apply the same inequality recursively, we have
$$\|q_0^\pi - q_0\|_{1,d_0^\pi} \leq \sum_{h=0}^{H-1} \|\mathcal{T}^\pi q_{h+1} - q_h\|_{1,d_h^\pi} \leq \sum_{h=0}^{H-1} \|\mathcal{T}^\pi q_{h+1} - q_h\|_{2,d_h^\pi}$$

The last inequality follows from the Jensen's inequality. Therefore, we have
$$|v^\pi(s_0) - \sum_a \pi(a|s_0)q_0(s_0, a)| \leq H\sqrt{C(\varepsilon_{opt} + \varepsilon_{apx} + c_0 H R_n^\mu(\mathcal{F}) + c_0 H^2 \sqrt{\frac{\log(H/\delta)}{n}})}$$
with probability $1 - \delta$.

The bound on FQE estimate holds for one policy in the candidate set. Using the union bound, the bound can hold for all policies. Therefore, the OPS algorithm has error up to
$$2H\sqrt{C(\varepsilon_{opt} + \varepsilon_{apx} + c_0 H R_n^\mu(\mathcal{F}) + c_0 H^2 \sqrt{\frac{\log(|\Pi|H/\delta)}{n}})}$$ with probability at least $1 - \delta$. $\qquad\square$

## A.5 DETAILS OF SECTION 4.2

By the performance difference lemma,
$$J(\pi^*) - J(\pi) = \mathbb{E}^{\pi^*}\left[\sum_{h=0}^{H-1} q_h^*(S_h, A_h) - q_h^*(S_h, \pi_h(S_h))\right]$$
$$\leq \mathbb{E}^{\pi^*}\left[\sum_{h=0}^{H-1} q_h^*(S_h, A_h) - q_h(S_h, A_h) + q_h(S_h, \pi_h(S_h)) - q^*(S_h, \pi_h(S_h))\right]$$
$$\leq \mathbb{E}^{\pi^*}\left[\sum_{h=0}^{H-1} |q_h^*(S_h, A_h) - q_h(S_h, A_h)| + |q_h(S_h, \pi_h(S_h)) - q^*(S_h, \pi_h(S_h))|\right]$$
$$\leq \sum_{h=0}^{H-1} \|q_h^* - q_h\|_{1,d_h^{\pi^*}\pi^*} + \|q_h^* - q_h\|_{1,d_h^{\pi^*}\pi_h}$$
$$\leq \sum_{h=0}^{H-1} \|q_h^* - q_h\|_{2,d_h^{\pi^*}\pi^*} + \|q_h^* - q_h\|_{2,d_h^{\pi^*}\pi_h}$$

where $d_h^\pi$ is the state-action distribution at horizon $h$ induced by policy $\pi$. The first inequality follows because $\pi_h$ is greedy with respect to $q_h$. We consider a state-action distribution $\beta_0$ that is induced by some policy, then
$$\|q_0^* - q_0\|_{2,\beta_0} \leq \|\mathcal{T}q_1^* - \mathcal{T}q_1 + \mathcal{T}q_1 - q_0\|_{2,\beta_0}$$
$$\leq \|\mathcal{T}q_1^* - \mathcal{T}q_1\|_{2,\beta_0} + \|\mathcal{T}q_1 - q_0\|_{2,\beta_0}$$
$$\leq \|q_1^* - q_1\|_{2,\beta_1} + C\|\mathcal{T}q_1 - q_0\|_{2,\mu_0}$$

where $\beta_1(s', a') = \sum_{s,a} \beta_0(s, a) P(s, a, s') \mathbb{I}\{a' = \arg\max_{a'' \in \mathcal{A}} (q^*(s', a'') - q_1(s', a''))^2\}$ is also induced by some policy. The first inequality follows by the fact that $q^*$ is the fixed point of the operator $\mathcal{T}$. We can recursively apply the same process for $\|q_h^* - q_h\|_{2,\beta_h}$, $h > 0$, and we can get

$$\|q_h^* - q_h\|_{2,\beta_h} \leq C \sum_{h}^{H-1} \|\mathcal{T} q_{h+1} - q_h\|_{2,\mu_h}.$$

Therefore,

$$\|q^* - q_0\|_{2,\beta_0} \leq CH \sum_{h}^{H-1} \|\mathcal{T} q_1 - q_0\|_{2,\mu_0}.$$

Now we show that

$$\|q_k - \mathcal{T} q_k\|_{2,\mu}^2 \leq \min_{i=1,\ldots,|\Pi|} \|q_i - \mathcal{T} q_i\|_{2,\mu}^2 + c_0 V_{max} \sqrt{\frac{\log(|\Pi| H/\delta)}{n}}$$

holds with probability at least $1 - \delta$ for some constant $c_0$.

*Proof.* Using Hoeffding's inequality and the union bound, with probability at least $1 - \delta$

$$\left| \frac{1}{n} \sum_{(s,a,r,s') \in D_h} |q_{i,h}(s, a) - r - \max_{a'} q_{i,h}(s', a')|^2 - \|q_{i,h} - \mathcal{T} q_{i,h+1}\|_{2,\mu_h}^2 \right| \leq c_1 B \sqrt{\frac{\log(H|\Pi|/\delta)}{n}}$$

for all candidate value function $q_i = (q_{i,1}, \ldots, q_{i,H-1})$, $i = 1, \ldots, |\Pi|$ and $h \in [H]$.

Let $k$ be the index with the lowest empirical Bellman error, that is,

$$k = \arg\min_{i=1,\ldots,|\Pi|} \frac{1}{nH} \sum_{h=0}^{H-1} \sum_{(s,a,r,s') \in D_h} |q_i(s, a) - r - \max_{a'} q_i(s', a')|^2.$$

Then

$$\|q_k - \mathcal{T} q_k\|_{2,\mu}^2 \leq \min_{i=1,\ldots,|\Pi|} \|q_i - \mathcal{T} q_i\|_{2,\mu}^2 + c_0 V_{max} \sqrt{\frac{\log(|\Pi| H/\delta)}{n}}.$$

$\square$

## B EXPERIMENTAL DETAILS

We provide the experimental details in this section.

**Generating candidate policies.** To generate a set of candidate policies, we run CQL with different hyperparameter configurations on a batch of data with 300 episodes collected with an $\varepsilon$-greedy policy with respect to the optimal policy where $\varepsilon = 0.4$. The hyperparameter configuration includes:

- Learning rate $\in \{0.001, 0.0003, 0.0001\}$
- Network hidden layer size $\in \{128, 256, 512\}$
- Regularization coefficient $\in \{1.0, 0.1, 0.01, 0.001, 0.0\}$
- Iterations of CQL $\in \{100, 200\}$

That is, we have 90 candidate policies for Acrobot.

For Cartpole, CQL with many hyperparameter configurations can generate an optimal policy (reaching a return of 200) so the selection is sufficiently easy that OPS algorithms using both FQE and BE achieve zero regret. In order to make the result more meaningful, we remove all the policies that are optimal from the candidate set. That results in 67 candidate policies for Cartpole.

**Generating offline data for OPS.** To generate data for offline policy selection, we use three different data distributions: (a) a data distribution collected by running the mixture of all candidate policies. As a result, the data distribution covers all candidate policies well; (b) a data distribution collected by running the mixture of all candidate policies and an $\varepsilon$-greedy optimal policy that provides more diverse trajectories than (a); (c) a data distribution collected by a deterministic optimal policy.

**To perform experiment with multiple runs.** To perform experiments with multiple runs, we fix the offline data and the candidate policies and only resample the offline data for OPS. This better reflects the theoretical result that the randomness is from resampling the data for an OPS algorithm. In our experiments, we use $5$ runs and report the average regret with one standard error. Since the variability across runs is not large, we find using $5$ runs is enough.

**Random selection baseline** We include a random selection baseline that randomly chooses $k$ policies given $k$. Since the random selection algorithm has very high variance within one run, we first compute the expected regret of random selection by performing 2000 times random selection within each run, and report the average regret over runs.

**Practical considerations for FQE implementation.** As mentioned in the paper, we fix the function class as a two layer neural network model with hidden size 256 and tune the optimization hyperparameters. We use the Adam optimizer with learning rate selected from the set $\{0.001, 0.0003, 0.0001, 0.00003\}$ for 200 epochs. We selected the hyperparameter configuration that resulted in a value function with the smallest RMSTD error evaluated on a separate validation dataset. To avoid selecting an uncovered policy, we assign a low estimate to policies when FQE diverges to a large value.

