# OpenReview forum: "When is Offline Hyperparameter Selection Feasible for Reinforcement Learning?"
_ICLR.cc/2023/Conference — Submitted to ICLR 2023_

### Official Review · Reviewer_DgEQ · 2022-10-20

**Confidence:** 3
**Correctness:** 3
**Technical Novelty And Significance:** 3
**Empirical Novelty And Significance:** 2
**Recommendation:** 6

**Clarity, Quality, Novelty And Reproducibility:**

Quality: Theorems seem to be fine (have checked the easiest two of the proofs so not entirely sure).

Novelty: The paper shows that OPS is bounded by OPE which hasn’t been shown before; so that’s a novel and important insight.

Reproducibility: The empirical results seem not to be easily reproducible as there is no code available and no mention of what implementations have been used for the different OPEs.

Clarity: Overall, this paper wasn’t easy for me to read, although having quite some knowledge in RL and hyperparameters. In fact, it would help tremendously, if the authors would provide more intuitions and exemplary practical implications early on.
In particular, I have some questions for understanding/regarding clarity:

* How to find out n (n >= 100 as the empirical results suggest for both FQE and BE as OPE? The more, the better, but for FQE it seems to increase for high n?)?
* Clarity Sec. 3.3: It took me way too long to figure out the example MDP, a visualization would help massively. Also, why is a_h selected via argmin of pi? Where does the ⅛ come from in Corollary 1?
* Clarity Sec. 4: It took me some time figuring out what F is (~until related work), it would be nice if it would be mentioned briefly.
Just to make sure that I got it right: the FQE estimate for J(pi) is the expected value at the initial state?
* Sec 6: Why do you think that BE performs better than FQE for expert data even though C is high? (I assume C is high because only one policy has been used to generate the data so candidate policies are not well-covered.)
* Clarity Figure 1+2: For comparability please use the same scale on the y-axis. Why is the regret for diverse data way lower than for expert and well-covered data? On which performance metrics is the regret computed? Why do you only use 5 different runs? How does the (complexity of the) environment influence the required number of samples? It looks like Acrobot needs more episodes than Cartpole (at least for top-5). Why is the regret so high in Cartpole for top-10?
* How different are the generated candidate policies / J(pi)? Because if they are very similar it does not really matter how good OPE or OPS are.
* Appendix A.1, the equation after (1): is this the correct epsilon?
* App A.2: A small figure for the MDP would also be nice, makes it clearer right at the beginning of what is happening.
Is the general assumption that V_max is known? If yes, is it stated in the main paper? Is the general assumption that the rewards are bounded, if yes, was this also stated before the appendix? How are they bounded?
* App. A.3: What is W_i?


**Strength And Weaknesses:**

### Strengths:
* Theoretical analysis of the required number of episodes to perform offline policy selection (OPS) with the help of offline policy evaluation (OPE)

### Weaknesses:
* [Motivation of Hyperparameter Optimization/Selection]
The story of OPS for HP selection can be made stronger or adjusted altogether. My questions are:
Is it important that the policies which should be selected from are generated from the same algorithm, just with different hyperparameters (HPs)? Because if not, then the motivation for the paper is way broader because this is relevant for general selection among policies. In this case I see the candidate policies with different examples just as an example which is easy to construct. Also, if I would want to use OPS now to select suitable HPs for offline RL, what do the results tell me in practice?

* What hyperparameters are meant in the conclusion? Is it important to set up the collection of the dataset well (but how)? Or is it in general important how to set the HPs of the RL algo? Because of the latter I think this paper hinted at it briefly (we can see a large regret between the best policy from the top-k policies and best from all candidate policies, but we don’t know how bad some HP configurations are).
* [Clarity] In fact, there are many open questions for me (see next text box) that need to be clarified.


**Summary Of The Paper:**

The paper tackles the question of when offline policy selection (OPS, selecting a well-performing policy in offline RL) can be performed efficiently. For this they determine bounds based on bounds by offline policy evaluation (OPE): OPS is lower-bounded by the number of samples/episodes required for OPE. The paper explores when and under which assumptions fitted Q evaluation (FQE) is suitable for OPS (when candidate policies are well-covered by the dataset) and when Bellman errors (BE) for OPS are suitable (empirically performs better than FQE with small sample size).

**Summary Of The Review:**

My main concerns are the motivation, the clarity issues, making the paper hard to read, and an insight for practical use.

---

> ### Author Response · Authors · 2022-11-15
> **Response to Reviewer DgEQ (1/2)**
>
> Thank you for your feedback. We've made adjustments to the clarity in a revision as you suggested.
>
> > My questions are: Is it important that the policies which should be selected from are generated from the same algorithm, just with different hyperparameters (HPs)?  Also, if I would want to use OPS now to select suitable HPs for offline RL, what do the results tell me in practice?
>
> The policy set can be generated from different algorithms and hyperparameter configurations. In fact, we discuss a practical use case that FQE can be used to choose between pessimistic offline RL algorithm or behavior cloning. We show that we do not need to consider specific conditions where one outperforms the other as discussed in a previous ICLR paper (Kumar et al. 2022), since these conditions are generally hard to verify in practice. We are able to select the better algorithm and hyperparameters by using FQE. Does this answer your concern about the practical insight of the paper?
>
> In this paper we aim to investigate when OPS is possible. The main result is that, under general conditions, OPS can not be sample efficient so HP selection can not be sample efficient too. This highlights the difficulty of HP selection and we shouldn’t expect that there exists a universal HP selection algorithm in practice. We further study two conditions where there exists a sample efficient OPS algorithm. For example, as mentioned earlier, if we are selecting HPs for a pessimistic offline RL algorithm or a behavior cloning algorithm, we should use FQE for OPS.
>
> > What hyperparameters are meant in the conclusion? Is it important to set up the collection of the dataset well (but how)? Or is it in general important how to set the HPs of the RL algo? Because of the latter I think this paper hinted at it briefly (we can see a large regret between the best policy from the top-k policies and best from all candidate policies, but we don’t know how bad some HP configurations are).
>
> One of our results shows that if we don’t make additional assumptions then OPS requires exponential sample complexity. In section 4 and 5, we further discuss assumptions such as (1) good data coverage or (2) a well-covered candidate set to enable sample efficient OPS. That is, it is important to use additional assumptions for policy selection, otherwise, OPS can be impossible with limited data.
>
> > How to find out n (n >= 100 as the empirical results suggest for both FQE and BE as OPE? The more, the better, but for FQE it seems to increase for high n?)?
>
> n is the number of episodes in the dataset. The main goal of the paper is to study the sample complexity of the OPE and OPS, that is, the smallest n such that we can perform OPE or OPS reliably.
>
> FQE benefits more from a higher n since FQE involves fitting a regression so it requires a high sample size (scales with the function class being used for the regression). On the other hand, estimating BE in deterministic environments can be very sample efficient.
>
> > Clarity Sec. 3.3: It took me way too long to figure out the example MDP, a visualization would help massively. Also, why is a_h selected via argmin of pi? Where does the ⅛ come from in Corollary 1?
>
> We added a figure (Figure 3 in the appendix) for the proofs. $a_h$ is selected via argmin of $\pi$ to ensure the probability of choosing $a_h$ conditional on $s_h$ in the data is less than 1/A. The restricted range of epsilon (i.e., $\sqrt{1/8}$) comes from the use of Bernoulli reward instead of Guassian reward used in previous work. The analysis and results are cleaner with the restricted range.
>
> > Clarity Sec. 4: It took me some time figuring out what F is (~until related work), it would be nice if it would be mentioned briefly.
>
> Sorry for the confusion. We’ve edited the paper to be more clear about $\mathcal{F}$.
>
> > Just to make sure that I got it right: the FQE estimate for J(pi) is the expected value at the initial state?
>
> Yes, $J(\pi)$ denotes the expected return of a policy $\pi$.
>
> > Sec 6: Why do you think that BE performs better than FQE for expert data even though C is high? (I assume C is high because only one policy has been used to generate the data so candidate policies are not well-covered.)
>
> C is high because a deterministic policy is used to collect the data so there are some state-action pairs that would not be covered by the deterministic policy no matter how many episodes we have in the dataset. We are not sure why BE performs better than FQE for expert data. However, we’ve added a random selection baseline suggested by Reviewer H7wj and the result shows that both BE and FQE underperform the random selection baseline on Acrobot, that is, both perform poorly under expert data.

---

> ### Author Response · Authors · 2022-11-15
> **Response to Reviewer DgEQ (2/2)**
>
> > Clarity Figure 1+2: For comparability please use the same scale on the y-axis. Why is the regret for diverse data way lower than for expert and well-covered data? On which performance metrics is the regret computed? Why do you only use 5 different runs? How does the (complexity of the) environment influence the required number of samples? It looks like Acrobot needs more episodes than Cartpole (at least for top-5). Why is the regret so high in Cartpole for top-10?
>
> We’ve tried plotting the figures with a shared y-axis, however, since the range of each subplot is quite different (especially figure 2), some figures would not be readable after scaling the y-axis.
>
> Diverse data provides better data coverage, as a result, the regret of OPS should be lower. The regret is the difference in the policy value for the top-k policies and the best candidate policy.
>
> We used 5 runs since it is sufficient to separate two baselines.
>
> The complexity of function class affects the sample size as shown by the analysis for FQE. The complexity of the environment affects the approximation error (how well a function class can approximate the Bellman operator). In our results, the regret is low in Cartpole (less than 10).
>
> > How different are the generated candidate policies J(pi)? Because if they are very similar it does not really matter how good OPE or OPS are.
>
> In our experiments, the candidate policies range from very poor (e.g., return of 10 in Cartpole) to nearly optimal (e.g., return of ~H in Cartpole). We’ve added a random selection baseline as suggested by Reviewer vxfJ. iIf the candidate set is similar, the random selection baseline should perform well.
>
> > Appendix A.1, the equation after (1): is this the correct epsilon?
>
> $\epsilon’$ is set to be $\epsilon/|\Pi|$.
>
> > App A.2: A small figure for the MDP would also be nice, makes it clearer right at the beginning of what is happening. Is the general assumption that V_max is known? If yes, is it stated in the main paper? Is the general assumption that the rewards are bounded, if yes, was this also stated before the appendix? How are they bounded?
>
> Yes, we assume the rewards are bounded by $[r_{min}, r_{max}]$ for some known constant $r_{min}, r_{max} \in\mathbb{RR}$. We’ve edited the paper to state this explicitly in the background section in the revision. As a result, $V_{max}$ is at most $r_{max} H$ which is known.
>
> > App. A.3: What is W_i?
>
> $W_i$ is defined in the IS estimator in the beginning of page 5.
>
> > My main concerns are the motivation, the clarity issues, making the paper hard to read, and an insight for practical use.
>
> Please let us know if we have not fully addressed your concerns.

---

> > ### Comment · Reviewer_DgEQ · 2022-11-17
> > **Increased Score**
> >
> > Hi,
> >
> > Thank you very much for your reply to my and the others' reviews.
> > I think I have now gotten a much better understanding of your paper and increased the score accordingly.
> > However, I would like to remark that the motivation was much better clarified in the replies compared to the motivations in the paper. So, I would like to ask you to further polish that part of the paper. I might be willing to further increase my score.

---

> > > ### Author Response · Authors · 2022-11-18
> > > **Thank you for your response**
> > >
> > > Thank you for reading our responses and updating your score accordingly.
> > >
> > > We have uploaded a revision to better reflect the motivation. In summary, we changed the title to be about “policy” selection instead of “hyperparameter” selection. We have also updated the introduction (in blue text) to emphasize that the motivation of the paper is broader than hyperparameter selection.
> > >
> > > Any other suggestions on how to improve the paper are welcome.

---

> > > > ### Comment · Reviewer_DgEQ · 2022-12-05
> > > > **Thanks**
> > > >
> > > > Thanks for the further clarification as part of the revision. Highly appreciated.
> > > > We, the reviewers, engaged in an internal discussion. Therefore, I will not change my review for the moment.

---

### Official Review · Reviewer_fS4W · 2022-10-24

**Confidence:** 4
**Correctness:** 4
**Technical Novelty And Significance:** 2
**Empirical Novelty And Significance:** 3
**Recommendation:** 5

**Clarity, Quality, Novelty And Reproducibility:**

Clarity: This paper is very clearly written.

Quality: Proof of upper bound or construction of hard examples for lower bounds are correct. Existing results in this paper are generally solid and well organized. My main complaint is the content of this paper is not enough.

Novelty: See Strength And Weaknesses.

Reproducibility: No code provided.

**Strength And Weaknesses:**

Strength:
1. This paper studies an important problem for deploying offline RL algorithms, for example in the case of hyper-parameter search or model selection.
2.  This paper provides a clear answer about the relationship between OPS and OPE, in terms of worse-case sample complexity. This is a nice result to shed the light on algorithm design and analysis.

Weakness:
1. A large part of the paper contains either existing results or a trivial extension of them. Section 3.1 may not be explicitly stated in previous work, but should be considered a well-known result in the RL area. Section 3.2's result is novel, but the main techniques in the construction exist in previous work. Section 3.3 is also not new in some sense, given the OPS <-> OPE reduction. IS is the minimax optimal OPE estimator is known (E.g. Wang et al. 2017, Optimal and Adaptive Off-policy Evaluation in Contextual Bandits.). Theorem 4 in Section 4 is based on the error bound of FQE. A very similar form of this bound, if not exactly the same, exists in many OPE or offline RL literature. The discussion on known results can be reduced to give more space to discuss new results.
2. I like the discussion at the end of sections 3.2 and 3.3 which explains that the worst-case analysis may not be informative enough in when policy set \Pi has a certain structure. However, this is not discussed enough later. In practice, the policies for OPS are not arbitrary most time but have a certain structure: a sequence of checkpoints during training NN, policies from different hyperparameters and may have a monotonically increasing off-policyness or OPE variance, policies from function classes with monotonically increasing capacity. To understand when is offline hyperparameter selection feasible in practical scenarios, I think it is important to analyze some policy set structure that has a connection to the practical OPS problem. Section 4 gives an assumption about the policy \Pi, but it is a strong and uniform assumption, but not about the relationships between policies.
3. There are many other OPE estimators besides FQE. It would increase the significance of the empirical study if authors could analyze more and their different impacts on the OPS problems.

**Summary Of The Paper:**

This paper studies some theoretical properties of the offline policy selection (OPS) problem. It shows that this problem's worst case (in the sense of query policy set) sample complexity is upper-bounded and lower-bounded by the sample complexity of off-policy evaluation (OPE). Thus these OPS and OPE have the same difficulty in the worst-case sense. Then this paper provides a polynomial error bound for OPS based on Fitted Q evaluation (FQE) and Bellman error when the policy set is well-covered.

**Summary Of The Review:**

This paper discusses an important problem and provides some clear theoretical results. However, the provided result is too limited to give new insight into this problem.

---

> ### Author Response · Authors · 2022-11-15
> **Response to Reviewer fS4W**
>
> Thank you for your comments. We would like to first address your concerns about the originality of the theoretical results.
>
> > A large part of the paper contains either existing results or a trivial extension of them.
>
> We agree some results are extensions of analysis for OPE. We included these results in the main paper for completeness since similar results for OPS have not been shown explicitly in any paper. For example, as far as we know, no paper has shown IS also achieves nearly minimax sample complexity for OPS under general conditions. Therefore, we think that these results for OPS, which may be expected for experts like yourself, are not well known in the community. For example, Paine et al. 2020 mentions several empirical results that could be explained by these theoretical results.
>
> The goal of the paper is to investigate a different problem setting which has not been investigated thoroughly. Inevitably we would use the existing results of several OPE methods. We do not believe this should be considered a weakness of the paper since our goal is not to improve the existing analysis to derive a shaper bound but to understand a different setting.
>
> > IS is the minimax optimal OPE estimator is known (E.g. Wang et al. 2017, Optimal and Adaptive Off-policy Evaluation in Contextual Bandits.).
>
> Wang et al. 2017 show that IS achieves minimax MSE in the contextual bandits setting. We show that IS achieves minimax sample complexity for the OPS problem in the reinforcement learning setting. We’ve added a discussion in the revision.
>
> > The discussion on known results can be reduced to give more space to discuss new results.
>
> We appreciate this comment and understand that Theorem 2 is our primary result of interest, while Theorem 4 and corollary 1 are more auxiliary. However, we thought they merited inclusion because of their supporting role in the overall messaging of this paper, more so than the theoretical novelty they offer. However,  we can de-emphasize these in the revision of this paper. For example, we can include these results in Section 4 in the Appendix and summarize their core messages in the body of the paper.
>
> > To understand when is offline hyperparameter selection feasible in practical scenarios, I think it is important to analyze some policy set structure that has a connection to the practical OPS problem.
>
> We emphatically agree with this suggestion but we consider this direction as a future work. Before investigating specific policy set structures such as checkpoint selection you mentioned, we want to answer whether OPS is feasible more generally, which may be of greater interest for the general audience or for practical use. This paper provides a clear negative answer that we must exploit data coverage and MDP conditions (which we also studied in the paper) or some policy set structure (which we consider as an important future work for specific applications) to be able to perform OPS well.
>
> > There are many other OPE estimators besides FQE. It would increase the significance of the empirical study if authors could analyze more and their different impacts on the OPS problems.
>
> The reason we chose to study IS, FQE and BE is that they are the most common methods for policy selection (e.g., Paine et al., 2020). Other OPE estimates such as doubly robust estimators are extensions of these basic OPE estimators, which requires additional assumptions to outperform the basic estimators. For example, the DR estimate outperforms IS with a consistent reward estimator as shown by Wang et al.
> > However, the provided result is too limited to give new insight into this problem.
>
> We respectfully disagree with this statement. We clearly demonstrate that OPS is not easier than OPE by providing a new lower bound, so more assumptions are needed to enable OPS. Though this result may be unsurprising for experts like yourself, we believe that a clear paper with solid evidence documenting this result is beneficial for the broader community. This paper opens up future research directions such as exploiting policy set structure or other OPE methods, however, we unfortunately cannot answer too many questions in one paper.

---

> ### Comment · Reviewer_fS4W · 2022-12-10
> **Thanks the authors' response**
>
> I have read the author's response, but there is no satisfying answer to most of my points in the initial review. I will not change my score.

---

### Official Review · Reviewer_vxfJ · 2022-10-24

**Confidence:** 4
**Correctness:** 3
**Technical Novelty And Significance:** 2
**Empirical Novelty And Significance:** 2
**Recommendation:** 5

**Clarity, Quality, Novelty And Reproducibility:**

The paper is overall well written. However, some of the descriptions should better contextualize the results in the related work.

**Strength And Weaknesses:**

Strengths
- This paper studies an important and timely hyperparameter-tuning problem for the offline RL community.
- The results of Theorem 2 suggest that OPS is as hard as OPE in the worst case.
- There is extensive discussion of possible OPS methods leveraging OPE for sample efficient results.


Weaknesses
- Many of the results presented do not appear to be particularly new. While they may not have been stated exactly in the same form as presented here, I believe they share too great an overlap to overlook. Here are a few papers which, with some modifications and combinations, should be able to produce the results:

Corollary 1 -> Xiao et al 2022 (which is already cited)

Theorem 3 -> Dudik et al 2011 (Doubly robust policy evaluation and learning), Thomas & Brunskill (Data-Efficient Off-Policy Policy Evaluation for Reinforcement Learning)

Theorem 4 -> Duan & Yang, 2020 (Minimax-Optimal Off-Policy Evaluation with Linear Function Approximation) + Duan et al 2021 (already cited)


Minor points:
- The validity of Bellman error for deterministic settings has been discussed in Zhang and Jiang (2021) as well. They also have experiments with it.
- BVFT (Xie & Jiang and Zhang & Jiang) actually does not require Q* in the set. It allows approximation error (which estimation error and approximation error of a model class can be absorbed into). There are some shortcomings of BVFT (such as slower rate, stronger distribution assumptions, etc), but its generality is very good.


**Summary Of The Paper:**

This paper presents an analysis of the problem of offline policy selection OPS. The study is motivated by the problem of hyperparameter tuning for offline RL algorithms, which is emerging as an important direction to understand in order to effectively deploy learned policies. One approach the authors study is OPS. While it is obvious that OPS is possible via OPE, the authors show, in a minimax sense, that OPS is as hard as OPE. Later, several sample complexity results are derived for OPS based on known OPE methods. Experiments are presented comparing FQE and a standard (biased) Bellman error selector

**Summary Of The Review:**

I believe this paper is studying an important problem that deserves attention in the community, and it has some interesting insights and discussions as well as a lot of potential. I just think that it unfortunately falls short of the technical depth necessary for publication. In light of this, I have some suggestions that I hope will strengthen the paper for future submissions.
- I would try to focus less on the presentation of results that are either mostly known or easily derivable given the existing literature. Examples: Theorem 3, 4 and Corollary 1. They are fine discussion topics, but I think it’s hard to argue them as original results and they detract from the real important results.
- The result of Theorem 2 and the experiments are the most interesting parts of the paper by far. In particular, Theorem 2 opens up a flood of interesting research questions. For example, what is the true sample complexity of selecting an ‘epsilon’-best policy from a small set? Pursuit of this direction might, for example, mirror the rich literature that has emerged on best-arm identification in bandits [1, 2], despite worst-case bounds painting a fairly pessimistic picture (as Theorem 2 does here).

1. Jamieson & Nowak. “Best-arm Identification Algorithms for Multi-Armed Bandits in the Fixed Confidence Setting.”
2. Kaufmann et al. “On the Complexity of Best-Arm Identification in Multi-Armed Bandit Models”

---

> ### Author Response · Authors · 2022-11-15
> **Response to Reviewer vxfJ**
>
> Thank you for your reasonable and in-depth review..
>
> We would like to first address your concerns about the originality of some of the theoretical results. We want to first clarify that Theorem 3, 4 and Corollary 1 are results for the OPS problem, not for the OPE problem. Even though the OPE for OPS reduction (upper bound) is straightforward, these results in OPS have not been explicitly stated in existing works.
>
> > I would try to focus less on the presentation of results that are either mostly known or easily derivable given the existing literature. Examples: Theorem 3, 4 and Corollary 1. They are fine discussion topics, but I think it’s hard to argue them as original results and they detract from the real important results.
>
> We appreciate this comment and understand that Theorem 2 (lower bound) is our primary result of interest, while Theorem 4 and corollary 1 are more auxiliary. However, we believe they merit inclusion because of their supporting role in the overall messaging of this paper, more so than for the theoretical novelty they offer. However, we can de-emphasize these in the revision of this paper. For example, we can include these results in Section 4 in the Appendix and summarize their core messages in the body of the paper.
>
> > Theorem 2 opens up a flood of interesting research questions. For example, what is the true sample complexity of selecting an ‘epsilon’-best policy from a small set? Pursuit of this direction might, for example, mirror the rich literature that has emerged on best-arm identification in bandits [1, 2], despite worst-case bounds painting a fairly pessimistic picture (as Theorem 2 does here).
>
> Thank you for pointing out the literature on best-arm identification. However, we want to clarify that our setting is a purely offline setting while the literature in bandits has been focusing on the online setting (exploration and exploitation tradeoff or pure exploration). Moreover, do you mind clarifying what "the true sample complexity" refers to? In our paper, we provide a lower bound on the sample complexity of selecting an "epsilon"-best policy with high probability, and a matching algorithm.
>
> >  I just think that it unfortunately falls short of the technical depth necessary for publication.
>
> We very much respect the reviewer’s opinion. That said, we would like to re-emphasize
> that our goal is to investigate a less studied problem setting, OPS, instead of providing a sharper analysis of OPE methods. In studying the OPS problem, we inevitably would use the existing results of several OPE methods and this is not meant to pad the paper with novel analysis of existing OPE methods. We believe that our paper is the first paper studying the difficulty of offline policy selection for reinforcement learning, and highlighting this problem setting as well as the connection to OPE is an important contribution. The results may seem straightforward to you (an expert), but that does not mean that they are not useful to the community.

---

### Official Review · Reviewer_H7wj · 2022-10-25

**Confidence:** 3
**Correctness:** 4
**Technical Novelty And Significance:** 2
**Empirical Novelty And Significance:** 2
**Recommendation:** 6

**Clarity, Quality, Novelty And Reproducibility:**

* The paper is well written, even if it proposes dense notations and masks a little bit the underlying assumptions . The setting could be made more clear
* The paper gives theoretical insights about techniques that people are using, and help to better understand the link between policy selection and OPE.
* Experiments are maybe too simple, and having more diverse settings and better baselines would be interesting
* It is reproducible

**Strength And Weaknesses:**

First of all, the topic attacked by the article is very relevant and interesting. Being able to have better insights into the validity of the different OPE approaches for policy selection is relevant and interesting to the community. As a remark, I am not fully familiar with the theoretical literature in this particular domain and I did not check all the proofs. The way the article is written makes the contributions clear, even if the notations are a little bit dense. The defined assumptions show that the paper is restricted to a particular family of problems where i) actions are discrete and ii) the number of policies to compare is not too big which explains why the positioning is focused on hyperparameters selection. As a consequence; the results are valid only if the set of policies is not too large and cannot be used for instance when trying to find a good architecture (the experiments are indeed made on a set of 90 and 67 policies). This is a limit of the contribution. Particularly, I would be happy to have a better discussion about how the results hold when the number of possible actions is growing (As far as I understand, the sample complexity is exponential w.r.t the number of actions for OPE). Being more clear on the assumptions and discussing this point would facilitate the reading.

In addition to the sample complexity bounds between policy selection and OPE, the authors study two particular OPE algorithms and allow us to better understand what is happening when using one of these approaches as a selection criterion. These insights are then evaluated on two concrete sets of experiments achieved in two different environments. The authors study 3 different dimensions: the value of 'k' in a top-k approach, the number of needed episodes, but also the distribution of these episodes. The results do not give any advantage between BE and FQE, even if FQE seems to be a better approach in many cases. As a drawback of the analysis, the environments provided here are very simple, with very few possible actions, and I would encourage the authors to re-iterate their experimental protocol over some other benchmarks. For instance, it may be interesting to take a look at some D4RL environments (after discretization of the actions for instance), or environments like mazes (e.g minigrid?) where one could expect to have more than 2 or 3 possible actions. An unclear aspect also is the performance of a completely random selection algorithm. Particularly, we do not know the distribution of regrets over the set of learned policies, so maybe a random selection with k=10 would produce similar results. Adding these results is simple but important.



**Summary Of The Paper:**

The article is a theoretical paper studying policy selection through off-policy evaluation. More precisely, the paper proposes 3 contributions: a) a theoretical analysis of the sample complexity of policy selection w.r.t the sample complexity of OPE, b) the derivation of sample complexity for Fitted Q-evaluation, and c) a similar derivation when using the Bellman error as a selection criterion. In addition, the article proposes a few experimental results to check if the provided bounds are 'valid' when facing concrete problems.



**Summary Of The Review:**

In conclusion, the paper proposes interesting insight into off-policy selection, even if it is restricted to a particular setting (discrete actions + few policies to evaluate). The theoretical results connecting selection and OPE make sense, and the (too small) experimental study is validating the differences identified in the two OPE approaches that are studied.

---

> ### Author Response · Authors · 2022-11-15
> **Response to Reviewer H7wj**
>
> Thank you for your feedback and positive score! We have upload a revision according to your comments on the dense notation and assumptions.
>
> > I would be happy to have a better discussion about how the results hold when the number of possible actions is growing (As far as I understand, the sample complexity is exponential w.r.t the number of actions for OPE).
>
> We want to clarify that all results hold for a finite action set and the sample complexity of OPE and OPS grows exponentially w.r.t. the horizon H (instead of A), which is known as “the curse of horizon”.
>
> > As a drawback of the analysis, the environments provided here are very simple, with very few possible actions, and I would encourage the authors to reiterate their experimental protocol over some other benchmarks.
>
> We agree the paper would be stronger with more experiments. However, we want to first clarify that the difficulty of OPS in general is due to the curse of the horizon, rather than having a large action space. In our experiments, we perform experiments in environments with a horizon of 200, which is large (even with two actions, 2^200 is very large). We are working on experiments on higher dimensional environments. However, we think the current experiment is enough due to the fact that they are already long horizon tasks.
>
> > An unclear aspect also is the performance of a completely random selection algorithm. Particularly, we do not know the distribution of regrets over the set of learned policies, so maybe a random selection with k=10 would produce similar results. Adding these results is simple but important.
>
> Thank you for this suggestion. We’ve added this random selection baseline in the revision.
>
> Please let us know if we don’t fully address your concerns. We would also like to thank the reviewer for their helpful response, which has helped us improve the paper.

---

### Decision · Program_Chairs · 2023-01-20

**Decision:**

Reject

**Justification For Why Not Higher Score:**

Consensus on the lack of depth in the theoretical contribution and the worst case analysis on common OPE algorithms does not provide useful insights.

**Justification For Why Not Lower Score:**

N/A

**Metareview: Summary, Strengths And Weaknesses:**

This paper studies the sample complexity of offline hyperparameter selection (OPS). It presents a reduction of OPS to offline policy evaluation (OPE) and vice verse, and thereby gives both an upper and lower bound of OPS using the sample complexity of OPE. It further applies a few common OPE methods to OPS and analyzes their worst case sample complexity. Empirical studies are conducted with simple experiments.

Strengths:
- Sound theoretical study on the sample complexity of OPS. This has been rarely done in prior works.
- It builds a formal connection between OPS and OPE and shows that OPS is not easier than OPE in the worst case. This could be interesting for the community.

Weaknesses:
- Multiple reviewers are concerned with the depth of the theoretical analysis. Most conclusions can be derived easier from existing results on OPE once the reduction is built. The main interesting theoretical contribution is Theorem 2 that gives the lower bound of OPS. Unfortunately, this result is not sufficiently expanded to provide useful insights and impacts the significance of Theorem 2.
- Although OPS is shown the have the same level of difficulty with OPE in the worst case, it could be easier in some particular scenarios. Exploring problem structure to discovery these problems could be particularly useful to the community.

I would encourage authors to take into account the reviews and following up discussion seriously and prepare a revision for re-submission.


**Summary Of Ac-Reviewer Meeting:**

I discussed with two reviewers in the meeting and contacts the other reviewers offline. There were a consensus from most reviewers that the main theoretical contribution was only in Theorem 2 on the lower bound and the other results were relatively easy to derive from existing literature. That lead to the question of the amount of novelty and significance of contribution. Unfortunately, the authors did not expand their analysis following Theorem 2.

A few more suggestions from reviewers to improve the paper:
- Either conduct more experiments to study empirically whether the OPS performance could be better than the worst case scenario in some particular setting, or expand the theoretical analysis from Theorem 2.
- It would be useful to identify / define some problem dependent setting where OPS would be more efficient
- Consider other metrics commonly used in OPS beside the regret, such as rank correlation.

While some reviewer considers the theoretical results were useful to the community, they had lower confidence on the significance of the theoretical contribution. Therefore, I suggested to reject the submission in its current form.